# Attenuation Imaging with Ultrasound as a Novel Evaluation Method for Liver Steatosis

**DOI:** 10.3390/jcm10050965

**Published:** 2021-03-02

**Authors:** Po-Ke Hsu, Li-Sha Wu, Hsu-Heng Yen, Hsiu Ping Huang, Yang-Yuan Chen, Pei-Yuan Su, Wei-Wen Su

**Affiliations:** 1Department of Gastroenterology, Changhua Christian Hospital, Changhua 50006, Taiwan; 180358@cch.org.tw (P.-K.H.); 91646@cch.org.tw (H.-H.Y.); 182972@cch.org.tw (H.P.H.); 27716@cch.org.tw (Y.-Y.C.); 111252@cch.org.tw (P.-Y.S.); 2Institute of Medicine, Chung Shan Medical University, Taichung 40201, Taiwan; 3Department of Ultrasound, Changhua Christian Hospital, Changhua 50006, Taiwan; 32881@cch.org.tw

**Keywords:** attenuation imaging, ultrasound, liver steatosis

## Abstract

In recent years, ultrasound attenuation imaging (ATI) has emerged as a new method to detect liver steatosis. However, thus far, no studies have confirmed the clinical utility of this technology. Using a retrospective database analysis of 28 patients with chronic liver disease who underwent ultrasound liver biopsy and ATI, we compared the presence and degree of steatosis measured by ATI with the results obtained through liver biopsy. The area under the receiver operating characteristic curve (AUROC) of the ATI for differentiating between normal and hepatic steatosis was 0.97 (95% confidence interval: 0.83–1.00). The AUROC of the ATI was 0.99 (95% confidence interval: 0.86–1.00) in grade ≥2 liver steatosis and 0.97 (95% confidence interval: 0.82–1.00) in grade 3. ATI showed good consistency and accuracy for the steatosis grading of liver biopsy. Therefore, ATI represents a novel diagnostic measurement to support the diagnosis of liver steatosis in non-invasive clinical practice.

## 1. Introduction

In clinical medicine, the diagnosis of fatty liver is very important to promoting the treatment of chronic liver disease [1]. Fatty liver is generally considered to be a reversible and benign disease; however, researchers are increasingly trying to clarify its role in the etiology of various liver diseases [2]. The existence of fatty liver is related to steatohepatitis and can develop into liver fibrosis, cirrhosis, and even end-stage liver disease [3,4]. In patients with chronic hepatitis C, fatty liver accelerates the process of fibrosis, adversely affects the sustained viral response rate of antiviral therapy, and can predict the development of hepatocellular carcinoma caused by fatty liver [5,6]. The evaluation of liver steatosis is also very important in the prognosis of liver transplantation donors, because follicular steatosis of the donor liver is related to the risk of the transplant [7].

Currently, the gold standard method for diagnosing and evaluating fatty liver is a liver biopsy. However, this method is invasive, can easily cause complications (such as pain, bleeding, and infection), and requires pathological reports, which may delay the results. In addition, due to the high incidence of steatosis, its benign course, and the lack of a clear association with liver enzyme changes, a liver biopsy can only be used for certain patients in need, such as non-alcoholic steatohepatitis, because it is an invasive test. Therefore, it cannot be repeated, resulting in the inability to continuously monitor changes in steatosis patients over a period of time.

Studies have shown that using an ultrasound scanner (such as FibroScan^®^ (Echosens, Paris, France)) together with controlled attenuation parameters (CAP) to measure ultrasound attenuation can quantitatively assess the severity of fatty liver. However, FibroScan^®^ is not an imaging device, and therefore B-mode ultrasound evaluation cannot be performed at the same time [8,9].

On the other hand, many studies have shown that magnetic resonance imaging (MRI) can also quantify liver steatosis, and the accuracy of diagnosis is also quite high compared with liver biopsy [10]. However, MRI is expensive and time-consuming, and its clinical use is limited to medical centers with funds.

In recent years, Canon Medical Systems introduced a new attenuation imaging (ATI) mode to the market as a new ultrasound technique for diagnosing steatosis [11]. ATI can adjust the area of interest to evaluated liver attenuation and quantitatively grade the liver steatosis through ultrasound.

In view of the tools currently available to detect liver steatosis, there is no accurate, harmless, and easy-to-use detection method, other than CAP, to assess fatty liver. Therefore, this study aims to study the accuracy of a new generation of detection tools: ATI, in evaluating liver steatosis, to provide a convenient and reliable method for future clinical screening and the treatment of liver steatosis.

## 2. Methods

From 1 January 2019 to 31 July 2019, we included 48 patients with chronic hepatitis who planned to undergo liver biopsy at Changhua Christian Hospital. These patients met the following study inclusion criteria: (1) 18 to 80 years old, (2) body mass index (BMI) less than 35 but greater than 17 kg/m^2^, and (3) signed informed consent. At the same time, the exclusion criteria were as follows: (1) malignant, including hepatocellular carcinoma (*n* = 2) and cholangiocarcinoma (n = 2); (2) chronic system diseases, such as coronary artery disease, chronic kidney disease, and chronic respiratory system disease (*n* = 11); and (3) drinking alcohol (*n* = 5). Finally, 28 patients underwent liver biopsy and ATI for analysis. As mentioned above, all participants signed an informed consent form for this study, which was reviewed and approved by the Institutional Review Committee of Changhua Christian Hospital (Institutional Review Board, IRB No. 191101) for liver steatosis analysis.

### 2.1. Liver Biopsy

An ultrasound-guided 16-gauge needle was used to take liver biopsy specimens. A liver biopsy is at least 10 mm, and six portal tracts were required to be sufficient for scoring. The liver biopsy specimens were fixed with formalin, embedded in paraffin, and stained with Masson’s trichrome. At the same time, reticulin staining was used to facilitate the histological evaluation. Two independent pathologists analyzed all the specimens. They did not know the clinical and experimental characteristics of the patients tested. The degree of steatosis was determined by the percentage of fat cells in the liver sample, which can be seen on the glass slide as follows: S0: <5%; S1: 5% to 33%; S2: 34% to 66%; and S3: more than 66%.

### 2.2. ATI Measurement Method

The ATI was determined from data obtained using TOSHIBA^®^i800 (Toshiba, Tokyo, Japan) ultrasound equipment and was manipulated by technicians who were not aware of the results of other reports. An inter-observer agreement was arranged for image analysis. ATI provides the function of quantifying and reducing the color code of liver decay factors. This may be due to changes in the liver composition (such as increased fat content), see Figure 1.The ATI value was defined as db/cm/MHz × 100. If the following conditions were met, the measurement was considered valid:(1)At least five valid data points collected.(2)The success rate was over 60%.(3)Every R^2^ value was 0.9 or greater, and the data points were recorded.(4)The interquartile range was less than 30% of the median ATI (Figure 2).

### 2.3. Statistical Analysis

We evaluated the normal distribution of the quantitative variables. The data is reported as mean with standard deviation, or median with interquartile range. The Kruskal–Wallis test and the subsequent Dan Bonferroni post-test were used to analyze the differences in the degree of ultrasound fatty liver among the four groups (S0, S1, S2, and S3). A *p*-value of less than 0.05 was considered to indicate statistical significance.

The receiver operating characteristic curve (ROC) was used to evaluate the diagnostic performance of each non-invasive model. We calculated the area under the ROC curve (AUROC) and the 95% confidence interval (CI) of the AUROC. Then, we used the De Long method to compare the same data to determine the AUROC values of different diagnostic criteria. To evaluate the feasibility of this measurement, we calculated the diagnostic predictive values (i.e., the sensitivity, specificity, positive predictive value, and negative predictive value). For statistical analysis, we used MedCalc software version 19.4.0 (MedCalc, Ostend, Belgium; https://www.medcalc.org; 1 January 2020).

## 3. Results

From January 2019 to July 2019, we included 48 patients with chronic liver disease who planned to undergo liver biopsy. Subsequently, 4 cases of malignant tumors, 11 cases of chronic diseases, and 5 cases of current drinking patients were excluded. In the end, 28 patients met the following analysis selection criteria (Figure 3).

Table 1 lists the basic characteristics of the study participants. The average age of the patients was 50.8 (±14.0) years, and there were 8 men (28.5%). The risk factors for liver steatosis were the baseline mean body mass index value (27.2 kg/m^2^) and mean waist circumference (89.4 cm). There were seven (25%) diabetic patients. In addition, the lipid profile was the average level of triglycerides (149.0 mg/dL), cholesterol level (188.2 mg/dL), high-density lipoprotein level (50.4 mg/dL), and low-density lipoprotein level (125.2 mg/dL). The distribution of steatosis patients was as follows: six S0 patients (21.4%), five S1 patients (17.8%), nine S2 patients (32.1%), and eight S3 patients (28.5%). The etiology distribution included seven patients with hepatitis B virus infection (25.0%), nine patients with hepatitis C virus infection (32.1%), seven patients with non-alcoholic steatohepatitis (25.0%), and five (17.8%) autoimmune hepatitis patients.

In patients with general chronic liver disease, this study mainly evaluated the ATI assessment methods for liver steatosis of varying stages. The median (interquartile range) values for ATI according to the liver steatosis grade were 67.5 (54.0–69.0) for S0, 72 (72.0–78.5) for S1, 82.0 (81.5–85.7) for S2, and 98 (89.5–102) for S3, and with trends correlated to the liver steatosis grading (*p* < 0.001, Jonckheere–Terpstra trend test), see Figure 4. ATI triggered a significant difference in liver steatosis grade between S0 and S2, S0 and S3, and S1 and S3 with *p* < 0.05 under post-hoc analysis.

We then analyzed the cutoff values of ATI to correctly predict steatosis. For this reason, we performed an AUROC plot analysis, including all study participants (*n* = 28) with different steatosis grades. The AUROC of the ATI according to liver steatosis grade with 0.97 (0.83–1.00) in S ≥ 1, 0.99 (0.86–1.00) in S ≥ 2, and 0.97 (0.82–1.00) in S3, see Figure 5 and Table 2. The sensitivity values that distinguished normal and liver steatosis (≥S1) were 100% for ATI. The positive predictive value (PPV) of ATI was 95%, and negative predictive value (NPV) of ATI was 100%, which all indicated a high accuracy for the diagnosis of liver steatosis. See Table 2.

## 4. Discussion

In this study, we found that the ATI values of the grades of mild and severe steatosis increased significantly with the increase in the grade of steatosis diagnosed by histology (*p* < 0.001, Jonckheere–Terpstra trend test). The AUROC value of ATI proves that it can effectively distinguish different degrees of fatty liver (S0, S1, S2, and S3), which shows that ATI is a reliable and accurate diagnostic method for liver steatosis.

To date, no studies have conducted ATI measurements for the diagnosis of liver steatosis. Recent studies suggested that CAP assessment via transient elastography (TE) can quantify the diagnosis of liver steatosis [12]. The CAP mode of TE is a non-image-based ultrasound technology that is able to measure the stiffness of tissues in real time and accurately [13]. Simultaneously, this technology can measure liver steatosis in CAP mode using M and XL probes; here, it revealed the AUROC for hepatic steatosis grades S1 or higher, S2 or higher, and S3 or higher to be 0.82 (95% CI: 0.77–0.88), 0.83 (95% CI: 0.77–0.88), and 0.89 (95% CI: 0.84–0.93) for the M probe and 0.88 (95% CI: 0.82–0.93); 0.92 (95% CI: 0.89–0.96), and 0.93 (95% CI: 0.89–0.97) for the XL probe [14].

Another report showed that using the CAP mode of TE to detect liver steatosis in patients with hepatitis-C-presented AUROC values of 0.80 (95% CI: 0.75–0.84) for S1 or higher, 0.86 (0.81–0.92) for S2 or higher, and 0.88 (0.73–1) for S3. CAP exhibited a good ability to differentiate steatosis grades (Obuchowski measure = 0.92) [15].

In the past ten years, non-invasive MRI has provided a rapid, safe, and quantitative assessment of hepatic steatosis [16]. A study of diffusion-weighted MRI (DWI) to evaluate liver steatosis on the apparent diffusion coefficient (ADC) of liver fibrosis in patients with hepatitis C virus (HCV) genotype-4-associated chronic hepatitis showed that hepatic steatosis should always be considered for detecting hepatic fibrosis in histopathology [17]. These studies show that MRI can quantify liver steatosis, regardless of whether there is also liver inflammation or fibrosis, and it can be used for a wide range of diffuse liver diseases [18].

However, MRI is expensive and the inspection process is time-consuming. It is not portable and fast, like CAP with Fibroscan or ATI with ultrasound. Therefore, in the world, MRI is currently rarely used for the quantitative screening of liver steatosis, only for research purposes. Compared to our study, MRI showed the sensitivity and specificity of the fat–water ratio in detecting fatty infiltration in grade 2 at 96% and 85%, respectively [19]; however, our study with ATI reported higher sensitivity at 100% and specificity at 90%.

ATI is a mode used to estimate the ultrasonic attenuation coefficient in tissues. The parameters of this mode are attached to the two-dimensional color map on a B-mode ultrasound image. As compared with CAP, ATI’s advantage is the existence of an ultrasonic inspection mode; therefore, there is no need to arrange for additional equipment for examinations.

Recent studies indicated that, when comparing the measurement results of proton-density fat fraction based on magnetic resonance imaging with CAP assessment results based on TE, the former is more effective than CAP in evaluating liver steatosis [20,21,22]. Notably, ATI had a similar diagnostic mechanism in this study [23,24,25].

We acknowledge that there are several limitations, including a small sample size, relatively obese patients, unblinded patients, a possible selection bias in screening patients, and other easily negligent and unobserved biases due to a non-randomized-controlled clinical trial. In addition, the limitations of this technology include (1) the TOSHIBA^®^i800 machine and the update to the latest software that are required to perform ATI and (2) the additional time for physician education and training. To further demonstrate the efficacy of ATI, more statistical analyses and experimental evidence are required in the future. The advantages of this study are (1) the first comparative study of ATI as a novel measurement, (2) a prospective study with blinded operators makes the reports more credible, and (3) this study demonstrated that ATI was reliable for the diagnosis of liver steatosis.

## 5. Conclusions

ATI is a new method and showed results that are highly concordant with those of liver biopsy in detecting steatosis. ATI is a more reliable and noninvasive method for evaluating liver steatosis. This research is a milestone for the quantitative diagnosis of liver steatosis, and ATI with ultrasound may replace CAP with Fibroscan^®^ in the future.

## Figures and Tables

**Figure 1 jcm-10-00965-f001:**
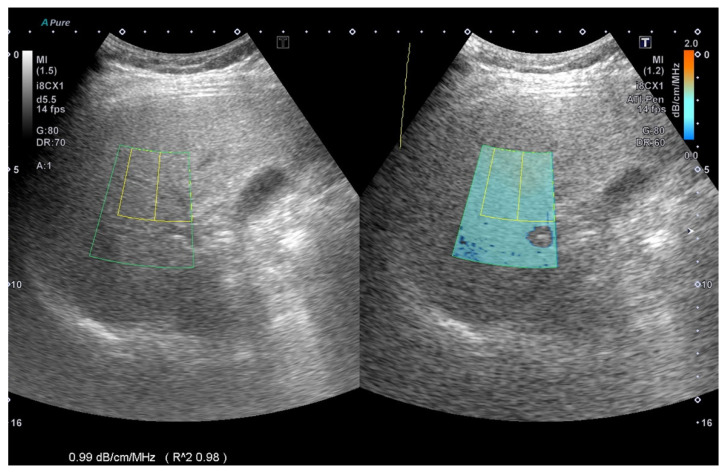
The measurement of ultrasound attenuation imaging. In this case, the ATI is 99 db/cm/MHz × 100 with R^2^: 0.98 as an effective value, ATI: attenuation imaging.

**Figure 2 jcm-10-00965-f002:**
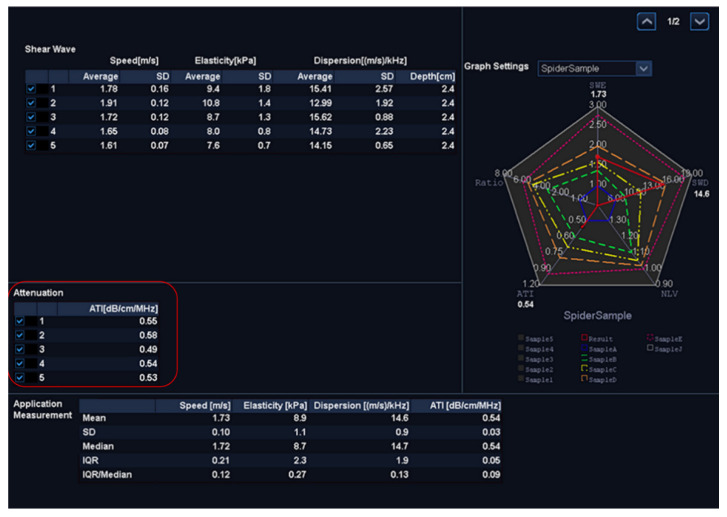
The median ATI from at least five valid data points (red box) with less than 30% of the IQR/median defined as the value of ATI. ATI: attenuation imaging, and IQR: interquartile range.

**Figure 3 jcm-10-00965-f003:**
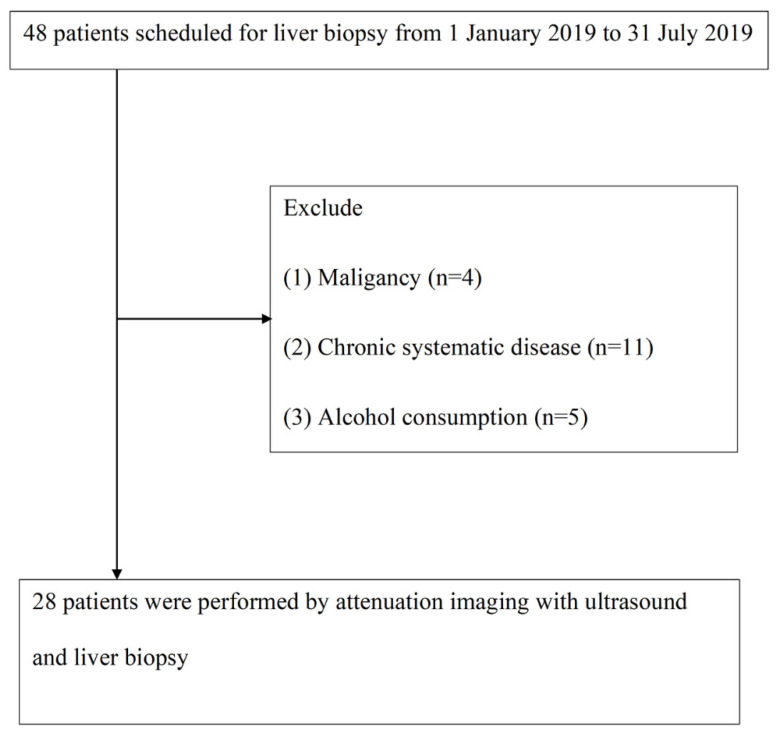
Study flowchart.

**Figure 4 jcm-10-00965-f004:**
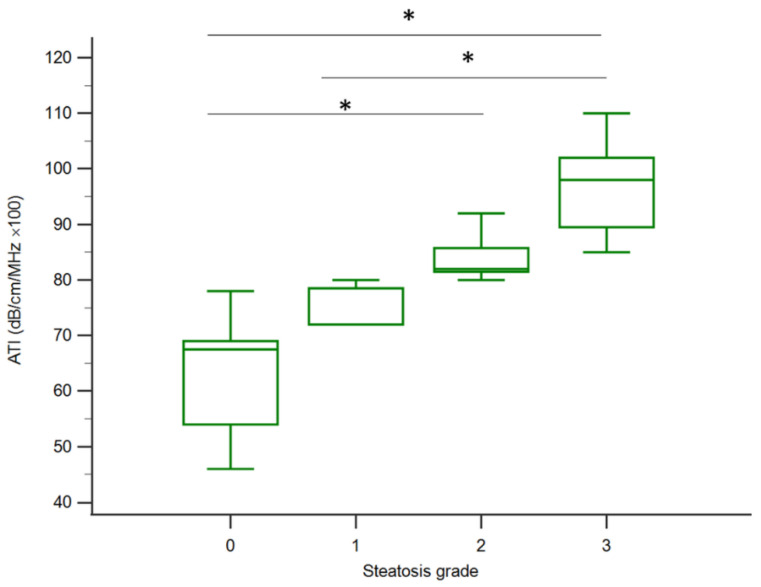
The distribution of ATI values according to the histologic steatosis grade. The ATI was significantly different between S0 and S2, S0 and S3, and S1 and S3 (*p* < 0.05) with the Kruskal–Wallis test under post-hoc analysis. Although S2 and S3 did not reach a statistical difference, the ATI in S3 still trended higher than S2. The vertical axis is a logarithmic scale. The top and bottom of the boxes are the first and third quartiles. The length of each box represents the interquartile range, within which are located 50% of the values. The lines through the middle of the boxes represent median values; ATI: attenuation imaging; * *p* < 0.05.

**Figure 5 jcm-10-00965-f005:**
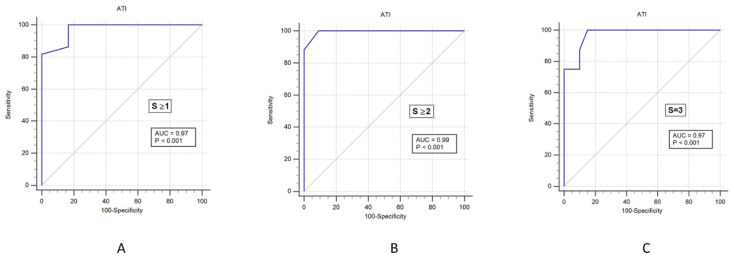
The AUROC of ATI for detecting different grades of liver steatosis. (**A**): Liver steatosis grade S1 or higher with AUROC values of 0.97 (95% confidence interval (CI): 0.83–1.00) for ATI. (**B**): Liver steatosis grade S2 or higher with AUROC values of 0.99 (95% CI: 0.86–1.00) for ATI. (**C**): Liver steatosis grade S3 with AUROC values of 0.97 (95% CI: 0.82–1.00) for ATI. AUROC: area under the receiver-operating characteristic curve; ATI: attenuation imaging.

**Table 1 jcm-10-00965-t001:** Clinical characteristics of the patients (*n* = 28).

Characteristic	Mean (±SD) or Absolute Count or Median (Interquartile Range)
Age, years	50.8 ± 14.08
Male sex, *n* (%)	8 (28.5)
BMI, kg/m^2^	27.2 ± 3.8
Waist	89.4 ± 12.8
Diabetes mellitus, *n* (%)	7 (25.0)
Triglyceride, mg/dL	149.0 ± 60.5
Cholesterol, mg/dL	188.2 ± 59.5
HDL, mg/dL	50.4 ± 20.2
LDL, mg/dL	125.2 ± 48.6
ALT, U/L	77.5 (34.3–125.8)
AST, U/L	56.5 (32.0–79.4)
Biopsy length, mm	16.5 ± 3.4
Liver stiffness, kPa	8.1 ± 2.3
Histology of steatosis grade *, *n* (%)	
S0	6 (21.4)
S1	5 (17.8)
S2	9 (32.1)
S3	8 (28.5)
Etiology of liver disease, *n* (%)	
HBV	7 (25.0)
HCV	9 (32.1)
NAFLD	7 (25.0)
AIH	5 (17.8)
ATI, dB/cm/MHz × 100	81.5 ± 14.1

AIH: autoimmune hepatitis; ALT: alanine transaminase; AST: aspartate transaminase; ATI: attenuation imaging; BMI: body mass index; HBV: hepatitis B virus; HCV: hepatitis C virus; HDL: high-density lipoprotein; LDL: low-density lipoprotein; NAFLD: nonalcoholic fatty liver disease; SD: standard deviation; * Steatosis grade; S0: <5%; S1: 5–33%; S2: 34–66%; and S3: >66%.

**Table 2 jcm-10-00965-t002:** ATI values for diagnosing liver steatosis.

Model	Steatosis Stage	AUROC (95% CI)	Cutoff	Sen	Spe	PPV	NPV
ATI	S ≥ 1	0.97 (0.83–1.00)	69	1.00	0.83	0.95	1.00
	S ≥ 2	0.99 (0.86–1.00)	78	1.00	0.90	0.94	1.00
	S = 3	0.97 (0.82–1.00)	82	1.00	0.85	0.72	1.00

ATI: attenuation imaging; AUROC: area under the receiver-operating characteristic curve; CAP: controlled attenuation parameter; CI: confidence interval; NPV, negative predictive value; PPV, positive predictive value; Sen, sensitivity; Spe, specificity; Cutoff values were obtained from the original article.

## Data Availability

Data available on request due to restrictions eg privacy or ethical.

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
