# Peer review of "Attenuation Imaging with Ultrasound as a Novel Evaluation Method for Liver Steatosis"

_jcm, 2021, doi:10.3390/jcm10050965_

Round 1
Reviewer 1 Report
More statistical analyses and experimental evidence is needed inorder to draw succinct conclusions with regard to the efficacy of the ATI. Institutional ethics committee approval details have not been detailed in the manuscript. The conceptual framework for using the ATI has not been detailed in the manuscript.
Author Response
Dear Editor,
We appreciate your editorial comments, as well as those of the reviewers, concerning our manuscript. Based on these comments, we have made several revisions to our manuscript, which is resubmitted for your consideration. If there is anything needing to be further improved, please do not hesitate to inform us at your earliest convenience. Your assistance is highly appreciated. We look forward to your message.
The followings are point-by-point responses to the comments.
Reviewer 1#:
More statistical analyses and experimental evidence is needed in order to draw succinct conclusions with regard to the efficacy of the ATI.
Response:
All authors thank the reviewer’s suggestions. We apologize for the statistical insufficient analysis of the efficiency of ATI with AUROC, NPV and PPV for fatty liver diagnosis. These may require a larger number of research samples and further experiments to prove. For this problem, we will explain and mention in the limitation of this study with text “In order to show the efficacy of ATI, more statistical analyses and experimental evidence are needed in the future. “in discussion with limitation, Line 214 to 215.
Thank you for your valuable comments. In the future, we may be able to conduct multivariate analysis or added on CAP for comparative research.
Reviewer 1#:
Institutional ethics committee approval details have not been detailed in the manuscript.
Response:
All authors appreciate the reviewer’s informative questions. We will add explanations and emphasis to the method paragraph "Institutional Review Board, IRB No. 191101 for liver steatosis analysis." And added on IRB certificate in supplement materials.in Line 70 to 71 in the manuscript.
Reviewer 1#:
The conceptual framework for using the ATI has not been detailed in the manuscript.
Response:
It is a great honor to give us this opportunity to correct.
We will added on Figure 2 for the explanation of the conceptual framework of ATI.
We have added all the knowledge about attenuation imaging with TOSHIBA®i800 (Toshiba, Tokyo, Japan) ultrasound in the appendix (supplementary materials).
The content in supplement materials contains:
- Images of actual operations in two cases.
- How to select the typed data and delete the deviation value.
- Add two links, one is toshiba's manual and the other is youtube introduction.
Thank you for your kind guidance.
Thank you for the opportunity to resubmit this manuscript for consideration of publication in Journal of clinical Medicine.
If you have any questions or comments regarding this manuscript, please do not hesitate to contact us by mail at our correspondence address, by fax at +886-4-7228289, by telephone at +886-4-7238595ext5501, or by e-mail at 180358@cch.org.tw or berger1912@gmail.com
Sincerely,
Po-Ke Hsu M.D
Department of Gastroenterology and Hepatology
Changhua Christian Hospital, Taiwan
Reviewer 2 Report
-Add the uniqueness of this study.
-Add shortly about the value of MR sequences in evaluation of fatty liver using these refs
-Besheer T, Razek AAKA, El Bendary M, et al. Does steatosis affect the performance of diffusion-weighted MRI values for fibrosis evaluation in patients with chronic hepatitis C genotype 4? Turk J Gastroenterol 2017;28:283-288.
-Image analysis by both observers with an inter-observer agreement
-Correct English language through the manuscript
-Compare your results with other studies discuss different ultrasound and MR sequences in the evaluation of fatty liver in the discussion section.
-Discuss the merits and limitations of the technique applied
-Update of references
Ref 19 is not in pubmed replace with another one
Author Response
Dear Editor,
We appreciate your editorial comments, as well as those of the reviewers, concerning our manuscript. Based on these comments, we have made several revisions to our manuscript, which is resubmitted for your consideration. If there is anything needing to be further improved, please do not hesitate to inform us at your earliest convenience. Your assistance is highly appreciated. We look forward to your message.
The followings are point-by-point responses to the comments.
Reviewer 2#:
Add the uniqueness of this study.
Response:
All authors thank the reviewer’s suggestions.
We have added two paragraphs to increase the uniqueness of this article:
“Furthermore, in the future, for the quantitative diagnosis liver steatosis, this research is a milestone, that is to say, perhaps ATI with ultrasound may replace CAP with Fibroscan.”
in conclusion paragraph, Line 222 to 224.
Reviewer 2#:
-Add shortly about the value of MR sequences in evaluation of fatty liver using these refs
Besheer T, Razek AAKA, El Bendary M, et al. Does steatosis affect the performance of diffusion-weighted MRI values for fibrosis evaluation in patients with chronic hepatitis C genotype 4? Turk J Gastroenterol 2017;28:283-288.
Response:
All authors appreciate the reviewer’s informative questions. This is indeed an important issue, and we have missed it. We further discuss the importance of MRI in liver steatosis.
We added on “In the past ten years, non-invasive MRI has provided a rapid, safe, and quantitative assessment of hepatic steatosis [16]. In the study of Diffusion-weighted MRI (DWI) to evaluate liver steatosis on the apparent diffusion coefficient (ADC) of liver fibrosis in patients with HCV genotype 4 associated chronic hepatitis showed hepatic steatosis should always be considered for detecting hepatic fibrosis in histopathology [17]. These studies show that MRI can quantify liver steatosis, regardless of whether there is also liver inflammation or fibrosis, it can be used for a wide range of diffuse liver diseases [18]. However, MRI is very expensive and the inspection process is very time-consuming. It is not portable and fast like CAP with fibroscan or ATI with ultra-sound. Therefore, in the world, MRI is currently rarely used for quantitative screening of liver steatosis, only for research purposes.” in discussion, Line 187 to 197.
Reviewer 2#:
Image analysis by both observers with an inter-observer agreement
Response:
All authors appreciate the reviewer’s informative questions. Indeed, we have an expert to do the final image analysis to reduce the observation bias, so we added "An inter-observer agreement is arranged for image analysis." in the method, Line 85.
Reviewer 2#:
-Correct English language through the manuscript
Response:
All authors thank the reviewer’s suggestions.
We will have our manuscript professionally edited before submission or read by a native English-speaking colleague. This can be carried out by MDPI's English editing service as the link:
https://www.mdpi.com/authors/english
Reviewer 2#:
Compare your results with other studies discuss different ultrasound and MR sequences in the evaluation of fatty liver in the discussion section
Response:
All authors thank the reviewer’s suggestions.
We added on the text with “Comparing to our study, MRI showed sensitivity and specificity of fat-water ratio to detect fatty infiltration in grade 2 were 96% and 85%, respectively, but our study with ATI reported more high sensitivity with 100% and specificity with 90%.” In discussion, Line 197 to 199
Reviewer 2#:
-Discuss the merits and limitations of the technique applied
Response:
Thank you for reminding this very important issue.
We added on the text “In addition, the limitations of this technology include (1) TOSHIBA®i800 machine and the update to the latest software are required to perform ATI (2) additional time for physician education and training.” In discussion of limitation in Line 212 to 215.
Reviewer 2#:
-Update of references
Ref 19 is not in pubmed replace with another one
Response:
All authors appreciate the reviewer’s informative questions.
We will delete this reference directly and update the reference.
Thank you for the opportunity to resubmit this manuscript for consideration of publication in Journal of clinical Medicine.
If you have any questions or comments regarding this manuscript, please do not hesitate to contact us by mail at our correspondence address, by fax at +886-4-7228289, by telephone at +886-4-7238595ext5501, or by e-mail at 180358@cch.org.tw or berger1912@gmail.com
Sincerely,
Po-Ke Hsu M.D
Department of Gastroenterology and Hepatology
Changhua Christian Hospital, Taiwan.

Reviewer 3 Report
Overall comments:
The work by Hsu et al. entitled ‘Attenuation imaging with ultrasound as a novel evaluation of liver steatosis’ is an important work that can be a milestone in this field. Without a randomized-controlled clinical trial, can it be concluded that ATI is a more reliable and non-invasive method for evaluating liver steatosis? In the conclusion, there should be a clear future direction for further research to establish the method. The discussion could be more details. In the discussion, the authors could compare with some other works even it could be for other diseases. For the transparency of the work, in the supplementary file, there could be all the patients imaging photo, if there is consent from the patients. The paper can be considered for publication only after the careful minor revision.
Specific comments:
Introduction, materials and methods and results are well presented.
Figure legends are independent. Therefore, it is mandatory to write details about what the authors mean. For example, in figure 3 there could be more details description.
Reference 19: The name should be in English.
There is some issue with the active and passive form.
English editing is required.
Author Response
Dear Editor,
We appreciate your editorial comments, as well as those of the reviewers, concerning our manuscript. Based on these comments, we have made several revisions to our manuscript, which is resubmitted for your consideration. If there is anything needing to be further improved, please do not hesitate to inform us at your earliest convenience. Your assistance is highly appreciated. We look forward to your message.
The followings are point-by-point responses to the comments.
Reviewer 3#:
Overall comments:
The work by Hsu et al. entitled ‘Attenuation imaging with ultrasound as a novel evaluation of liver steatosis’ is an important work that can be a milestone in this field. Without a randomized-controlled clinical trial, can it be concluded that ATI is a more reliable and non-invasive method for evaluating liver steatosis? In the conclusion, there should be a clear future direction for further research to establish the method. The discussion could be more details. In the discussion, the authors could compare with some other works even it could be for other diseases. For the transparency of the work, in the supplementary file, there could be all the patients imaging photo, if there is consent from the patients. The paper can be considered for publication only after the careful minor revision.
Response:
Our researchers are very pleased that you have a deep understanding of our research. The following points are our modifications and ideas. Thank you for your advice.
- Without a randomized-controlled clinical trial, can it be concluded that ATI is a more reliable and non-invasive method for evaluating liver steatosis?
Response :
We are just proposing a new technology, although there are almost no relevant clinical studies. The current data proves reliable and convenient, but the number of samples is still small and non-random research. We deeply feel that this research still has a long way to go. Our Changhua Christian Hospital has three machines: TOSHIBAI800, Fibroscan and MRI. We deeply hope that this arduous task will be completed in the near future. Maybe ATI was a superstar at that time.
2. In the conclusion, there should be a clear future direction for further research to establish the method. The discussion could be more details. In the discussion, the authors could compare with some other works even it could be for other diseases.
Response :
This is a very constructive suggestion. We are all grateful.
Reviewer 2# also gave us the same view, so we discussed and compared the sensitivity and specificity of MRI to liver steatosis, and added research reports related to MRI to supplement the deficiencies of our article. We are also very grateful your reminders and guidance.
Related changes are in Line 187 to 199 and 212 to 215.
3.For the transparency of the work, in the supplementary file, there could be all the patients imaging photo, if there is consent from the patients.
Response :
Thank you for your suggestion.
However, only two people agreed to supplement the visa, so we added two case reports to the supplementary information. However, the others were still unable to be contacted, so in order to show openness and transparency, we provide all research data.
- Case report in supplementary materials
- Excel raw data was added on
Finally, thank you for your understanding, in the future research, this kind of image data will be signed together with informed consent.
Reviewer 3#:
Specific comments:
Introduction, materials and methods and results are well presented.
Response:
Thank you for reading our research carefully, we are grateful.
Figure legends are independent. Therefore, it is mandatory to write details about what the authors mean. For example, in figure 3 there could be more details description.
Response:
Thank you for your valuable suggestions
We added Figure 1, Figure 2, and Figure 3 legends to make the figure easier to read.
Reference 19: The name should be in English.
Response:
I am very sorry, as the reviewer said, because Pubmed cannot be found, only Google Scholar has it. Because it is not representative, I deleted the reference and replaced by extended discussion in Line 189 to 201 and 214 to 217.
There is some issue with the active and passive form.
English editing is required.
Response:
All authors thank the reviewer’s suggestions.
We will have our manuscript professionally edited before submission or read by a native English-speaking colleague. This can be carried out by MDPI's English editing service as the link: https://www.mdpi.com/authors/english.
Thank you for the opportunity to resubmit this manuscript for consideration of publication in Journal of clinical Medicine.
If you have any questions or comments regarding this manuscript, please do not hesitate to contact us by mail at our correspondence address, by fax at +886-4-7228289, by telephone at +886-4-7238595ext5501, or by e-mail at 180358@cch.org.tw or berger1912@gmail.com
Sincerely,
Po-Ke Hsu M.D
Department of Gastroenterology and Hepatology
Changhua Christian Hospital, Taiwan.
